# Open Microfluidic Cell Culture in Hydrogels Enabled by 3D-Printed Molds

**DOI:** 10.3390/bioengineering12020102

**Published:** 2025-01-22

**Authors:** Madison O’Brien, Ashley N. Spirrison, Melati S. Abdul Halim, Yulai Li, Adrian Neild, Catherine Gemrich, Reza Nosrati, Luis Solorio, Max M. Gong

**Affiliations:** 1Jim and Joan Bock Department of Biomedical Engineering, Trine University, Angola, IN 46703, USA; howar366@purdue.edu (M.O.); ashley.n.spirrison@vanderbilt.edu (A.N.S.); 2Weldon School of Biomedical Engineering, Purdue University, West Lafayette, IN 47906, USA; cgemrich@purdue.edu (C.G.); lsolorio@purdue.edu (L.S.); 3Department of Mechanical and Aerospace Engineering, Monash University, Melbourne, VIC 3168, Australia; melati.abdulhalim@monash.edu (M.S.A.H.); ylii0235@student.monash.edu (Y.L.); adrian.neild@monash.edu (A.N.); reza.nosrati@monash.edu (R.N.)

**Keywords:** open microfluidics, hydrogel, collagen, 3D-printed molds, blood vessel mimics

## Abstract

Cell culture models with tissue-mimicking architecture enable thein vitro investigation of cellular behavior and cell–cell interactions. These models can recapitulate the structure and function of physiological systems and can be leveraged to elucidate mechanisms of disease. In this work, we developed a method to create open microfluidic cell cultures in vitro using 3D-printed molds. The method improves sample accessibility, is simpler to manufacture than traditional closed microfluidic cell culture systems and requires minimal specialized equipment, making it an attractive method for cell culture applications. Further, these molds can generate multiple tissue-mimicking structures in various hydrogels, including blood vessel mimics using endothelial cells (HUVECs). Various geometries were patterned into agarose, gelatin, and collagen type I hydrogels, including star-shaped wells, square wells, round wells, and open channels, to demonstrate the versatility of the approach. Open channels were created in collagen with diameters ranging from 400 µm to 4 mm and in multiple collagen densities ranging from 2 mg/mL to 4 mg/mL. To demonstrate the applicability of our approach for tissue modeling, blood vessel mimics were generated in open channels with diameters of 800 µm and 2 mm, with high cell viability (>89%) for both dimensions. The vessel mimics were used to study the effects of hypoxia on cell viability and CD31 expression by subjecting them to a reduced-O2 environment (∼16% $O_{2}$). As compared to normoxia conditions, vessel mimics under hypoxia had a reduction in cell viability by 8.3% and CD31 surface expression by 7.4%. Overall, our method enables the generation of different geometries in hydrogels and the development of in vitro tissue mimics for biological applications.

## 1. Introduction

Cell culture is the cornerstone of the development of tissue models for biomedical applications. Conventional 2D cell culture methods involve growing and maintaining cells on flat surfaces (e.g., Petri dishes, flasks, and well-plates). While these approaches are widely used due to their simplicity and low cost, 2D cell culture techniques fail to recapitulate the in vivo structure and microenvironment of tissues. Consequently, the advancement of 3D cell culture systems, such as spheroids and organoids, has allowed researchers to better mimic in vivo tissue structure and function [1,2]. Similarly, the development of microfluidic cell culture systems (i.e., organs-on-chips) has dramatically improved the capability to model tissues and their microenvironments at relevant cellular scales [3,4].

Microfluidics has been integral to the development and innovation of new technologies for cell culture [3]. These small-scale technologies reduce the reagent volume, cost, and time required to perform reactions, making them applicable to a variety of biological applications, ranging from drug screening to cancer modeling and beyond [5]. Moreover, microfluidic cell culture approaches aim to improve upon conventional cell culture methods by allowing for the study of low cell densities at a high spatial and/or temporal resolution, while simultaneously controlling the local cellular microenvironment [6,7]. The first microfluidic cell culture devices utilized fully closed channels, where fluids are pumped into an enclosed system with no air–liquid interface. Cells can be seeded into these channels with flowing media, which is more representative of the dynamic environments seen in vivo [8]. Closed microfluidic systems have been utilized to better mimic normal and pathological tissue structure, including the modeling of blood vasculature and lymphatic vessels [9,10].

The field of microfluidics quickly evolved to include other device designs, including open microfluidics, where the fluid path is not bound by a channel and has at least one air–liquid interface [11]. Examples of open microfluidic systems range from simple droplets on surfaces to more complex capillary-driven open channels [12,13,14,15,16]. In contrast to closed microfluidic systems, open microfluidic devices negate the need for multistep fabrication and for specialized equipment to drive fluid flow. These advantages increase reproducibility between devices during fabrication and reduce the issue of trapped air bubbles often seen in closed microfluidic devices [14,17,18,19,20,21]. In the context of cell culture, open microfluidic channels allow for larger cellular samples, such as spheroids, to be used in the system without constraint or damage to the structure of the sample [22]. Monitoring of the media properties (i.e., pH) using external probes, the collection of media for conditioned media analysis, and interaction with surface assays are also more accessible in open microfluidic systems [22,23,24].

In this work, open microfluidic cell culture systems were developed using 3D-printed molds to improve the accessibility of fabricating tissue-mimicking architecture in hydrogels for in vitro modeling. Our method enables the formation of various architectures in agarose, gelatin, and collagen hydrogels. Star-shaped wells, square wells, and round wells were patterned into agarose and gelatin to demonstrate the versatility of the approach for generating simple and complex geometries in different gels. Open channels were formed in collagen type I gel for the creation of blood vessel mimics. The diameters of channels successfully molded into the collagen ranged from 400 µm to 4 mm. Open blood vessel mimics were formed by seeding human umbilical vein endothelial cells (HUVECs) into the 800 µm and 2 mm diameter channels, with high viability and relevant morphology. To demonstrate the applicability of our approach for biomedical applications, we investigated the effects of hypoxia on the blood vessel mimics using 16% oxygen ($O_{2}$). The vessel mimics showed reduced viability and CD31 expression under hypoxia when compared to normoxia conditions. Overall, our approach enables the generation of open structures in different hydrogels that enable the development of tissue mimics for biomedical applications.

## 2. Materials and Methods

### 2.1. Three-Dimensionally-Printed Mold Fabrication

SolidWorks was used to design all 3D-printed molds. These molds were dimensioned to fit into an individual well of a 24-well multiwell plate to facilitate the ease-of-use and culturing of tissues in standard cell cultureware. Each mold consisted of a base piece with the pattern to be imparted into the chosen hydrogel and a post piece for stabilization (Figure 1A). The base of each mold had crescent-shaped cutouts on both sides to allow hydrogel to be dispensed homogeneously around the pattern and to reduce air bubble formation.

Different pattern geometries were fabricated, including a star-shaped well, a square well, and a round well, to demonstrate the generation of shapes with varying radii and angles (Figure 1D). Open channels, consisting of a semi-cylindrical geometry, were also designed to create open blood vessel mimics.

The star-shaped, square, and round well molds were printed from PLA filament (SUNLU, Irvine, CA, USA) using a Prusa i3 MK3S+ printer (Prusa Research, Prague, Czech Republic). The open-channel molds were printed from VeroBlue resin (Stratasys, Eden Prarie, MN, USA, RGD840) using an Eden260VS (Stratasys, Eden Prarie, MN, USA) printer with accuracy between 20 and 85 µm and a minimum layer height of 16 µm. Both PLA filament and VeroBlue resin were used in this work because they have distinct advantages of lower costs and higher resolution, respectively. PLA and VeroBlue resin both proved to have sufficient material properties for this hydrogel molding application. Ultimately, we used a Stratasys resin printer to create higher-resolution molds for patterning the open channels for the blood vessel mimics. The base and post pieces of each mold were assembled using an adhesive. All molds were sterilized under UV light for 30 min before use. All molds were imaged using a digital microscope (Hirox-USA, inc., Oradell, NJ, USA, KH-8700).

### 2.2. Agarose and Gelatin Hydrogel Preparation and Loading into Multiwell Plate

Agarose powder (Fisher Bioreagents, Waltham, MA, USA, BP160) was added to 1x phosphate-buffered saline (PBS, Sigma Aldrich, St. Louis, MO, USA, P5493) solution to make 1% and 2.5% gel solutions. Gelatin Type A from porcine skin (Sigma-Aldrich, St. Louis, MO, USA, G2500) was added to 1x PBS solution to make 5% and 10% gel solutions. Each solution was microwaved for one minute to fully dissolve the powder. The assembled molds were placed into each well of the prepared well plate, and molten agarose or gelatin (300 µL) was dispensed through the crescent-shaped opening on one side of the mold using a pipette until the entire cavity beneath the mold was completely filled. Each hydrogel was allowed to gel at 4 °C for 20 min before molds were removed to reveal the patterned architecture for imaging. No additional crosslinking techniques were used.

### 2.3. Collagen Gel Preparation and Loading into Multiwell Plate

Collagen hydrogel was generated using rat tail collagen I (Corning, Corning, NY, USA, 354249). Stock solution was diluted with 5x PBS and pH-balanced with 0.5 M sodium hydroxide (NaOH, Sigma Aldrich, St. Louis, MO, USA, 58310C) to achieve a final concentration of 4 mg/mL collagen in 1x PBS with a pH of 7.4. Other concentrations of collagen were made by adjusting the volume of the stock collagen solution, PBS buffer, and NaOH neutralizing solution. All collagen preparation was performed on ice to slow polymerization before loading. Collagen (300 µL) was pipetted into the prepared well plate with molds as described in Section 2.3. Collagen was allowed to gel at room temperature for 15 min before being transferred to 37 °C for 1 h to finish the gelation process. No additional crosslinking techniques were used. Molds were removed after the 1 h incubation period to reveal gelled collagen containing the desired molded architecture. All collagen preparation and loading steps were performed in a sterile biosafety cabinet.

### 2.4. Cell Culture

Human umbilical vein endothelial cells (HUVECs, ScienCell, Carlsbad, CA, USA, 8000) were cultured at 37 °C in standard T75 culture flasks at a seeding density of 600,000 cells. Cultures were maintained with the PromoCell Endothelial Cell Growth Medium MV 2 (MV 2, PromoCell, Heidelberg, Germany, C-22221) with an endothelial supplement pack (PromoCell, Heidelberg, Germany, C-39221), 5% fetal bovine serum (FBS, Gibco, Waltham, MA, USA, 12676029), and 1% Penicillin Streptomycin (Cyvita, Marlborough, MA, USA, SV30010). HUVECs were cultured until 80% confluency was reached, and only cells under passage 10 were used. All cultures were kept in a humidified incubator at 37 °C with 5% CO_2_.

### 2.5. Open-Channel Cell Culture for Blood Vessel Mimics

HUVECs (20 µL) were loaded into open channels patterned in collagen, as prepared using the steps in Section 2.4, at a seeding density of 60,000 cells/mL. This high seeding density was chosen based on previous work by Gong et al., but adapted for an open channel rather than a full lumen [10]. This density proved to be sufficient in covering the entire surface area of the open channel. The culture was placed on an incubated lab shaker for five minutes at 100 rpm to ensure all the cells were in the channel, then incubated at 37 °C for one hour to allow the cells to adhere to the collagen before MV-2 media was added. Media changes were performed daily to maintain the cells. The cultures were checked daily and were fixed with 4% PFA after five days once cells covered the entire surface area of the open channel. For the hypoxia study, two 24-well multiwell plates of blood vessel mimics were prepared as described above. After 24 h of incubation, one plate of vessels was moved to 16% oxygen ($O_{2}$, hypoxic condition), while the second plate was left at 20% $O_{2}$ (normoxic condition). Media changes were performed daily, and the vessels were fixed after four days for image analysis.

### 2.6. Immunofluorescence Staining and Imaging

The cytoskeleton, nucleus, and CD31 of the endothelial cells in the blood vessel mimics were immunostained to visualize vessel structure and cell coverage. The vessels were fixed in 4% paraformaldehyde (PFA, Thermo Scientific, Waltman, MA, USA, J19943K2) for 10 min before rinsing three times with 1x PBS. Cell membrane permeability was increased by adding 0.1% Triton-X solution (Sigma Aldrich, St. Louis, MO, USA, X100) for five minutes, followed by three washes with 1x PBS. The fixed vessels were blocked with 1% bovine serum albumin (BSA, Sigma Aldrich, St. Louis, MO, USA, A9647) in 1x PBS for one hour before washing 3x with PBS.

CD31 monoclonal antibody (Thermo Scientific, Waltman, MA, USA, MA516337) was diluted 1:50 in 1x PBS and added to the vessels overnight. The culture was placed on a lab shaker and kept in a cold room at 4 °C. The samples were washed three times with 1x PBS before adding a 5 µg/mL goat IgG anti-rabbit Alexa Fluor 568 secondary antibody (Invitrogen, Waltman, MA, USA, A-11011) for one hour on a lab shaker at room temperature. The vessels were washed three times with 1x PBS before adding a 1x solution of Phalloidin AlexaFluor 488 (Thermo Fisher, Waltman, MA, USA, A12379) for 30 min at room temperature. After three 1x PBS washes, a 5 µg/mL DAPI solution (Invitrogen, Waltman, MA, USA, D21490) was added for 10 min at room temperature. The vessels were washed three times with 1x PBS and left in the last wash for imaging. Imaging was performed using a Zeiss 880 inverted confocal microscope (Ziess Microscopy, Jena, Germany) at 10x magnification. All image analyses were performed in ImageJ Fiji 2. CD31 surface area coverage was calculated using thresholding in Fiji.

### 2.7. Cell Viability Analysis

To assess the viability of the blood vessel mimics, a 1:1 dilution of MV-2 cell medium and Vitastain Acridine Orange/Propidium Iodide (AO/PI) solution (Perkin Elmer, Waltman, MA, USA, NC2019706) was prepared. The solution was pipetted into each vessel and allowed to incubate for 10 min at room temperature before confocal imaging.

A MATLAB script was used to obtain live/dead counts for AO/PI-stained blood vessel mimics. The program batch-processed multiple image files and returned the live/dead counts and cell viability for each sample. The program incorporated both binary masking and edge-finding techniques to distinguish cells from the background. For each Carl Ziess Image (.czi) file, the program split the channels into files to separate the live/dead cell images. For each channel, the program increased the contrast of a binary version of the image and then used edge-finding to outline all fluorescent cells. The program then counted the objects that were outlined. Threshold values were set for both live and dead images to ensure objects below a certain size were not included in the count, reducing noise and errors. The program was validated with manually counted live/dead images prior to data analysis.

### 2.8. Statistical Analysis

One-way analysis of variance (ANOVA) was utilized to compare the mean values of the cross-sectional area between open channels molded at varying collagen densities. Dunnett’s multiple comparisons test found that all collagen densities were not significantly different from the control value (cross-sectional area of the mold), and they were therefore utilized for further studies (Figure 2C). Multiple unpaired *t*-tests were utilized to determine whether the mid-plane and cross-sectional area measurements of the open channels (Figure 2D,E) were statistically different from the mold dimensions. Unpaired t-tests were also used to compare cell viability and CD31 surface coverage between blood vessel mimics in hypoxia versus normoxia conditions (Figure 4B,D). The tests utilized the Holm–Šídák method and assumed individual variance for each row. For each experiment, a minimum sample size of *n* = 3 was used.

## 3. Results

### 3.1. Generating Hydrogel Models Using 3D-Printed Molds

To demonstrate the generation of models with tissue-mimicking architecture, we leveraged 3D-printed molds to pattern customized geometries in different hydrogels, as illustrated in Figure 1. The current approach allows a high degree of customizability in terms of the structure to be molded into the hydrogels, such as a star-shaped well, square well, and round well (Figure 1A). The designs for the molds were first created in computer-aided design (CAD) software (Solidworks 2021), and then prototyped via 3D printing (Figure 1B). An example of the molding process is depicted in Figure 1C using the 3D-printed open-channel mold. The mold was placed in the well of a 24-well multiwell plate, which was subsequently filled with collagen type I gel. Unmolded collagen was compared to molded collagen, where the curvature of the open channel was visualized through the addition of dyed liquid. This qualitative assessment demonstrates that the molding process is feasible for creating the external contours of the geometry on the 3D-printed mold.

### 3.2. Generation of Various Architectures in Agarose and Gelatin

Agarose and gelatin have been previously used for various cell culture applications. Agarose has been used at a concentration of 1% (*w*/*v*) to create spheroid cultures or at 2.5% (*w*/*v*) for single-cell micropatterned cultures [25]. Gelatin, or gelatin derivatives such as gelatin methacryloyl (GelMA), has been used to fabricate microdevices for cell culture purposes at 5% (*w*/*v*) [26] or 10% (*w*/*v*) [27,28]. To demonstrate that our approach could be used to mold different architectures in a variety of hydrogels used for cell culture, we used star-shaped, square, and round well molds to pattern agarose and gelatin into their respective shapes. Top view images of the patterned shapes in 1% agarose, 2.5% agarose, 5% gelatin, and 10% gelatin showed structurally intact star, square, and semi-spherical wells (Figure 2). The hydrogels used for this method needed to possess physical properties capable of producing structurally stable architectures to ensure they were suitable for the desired cell culture application. All structures were fabricated in both hydrogels, and the structures remained intact after mold removal regardless of the concentration or desired shape. These results indicate that 3D-printed molds can be used to create tailored geometries in commonly used cell culture hydrogels.

### 3.3. Characterization of Open Channels in Collagen

Open-channel molds of different diameters (400 µm, 800 µm, 1 mm, 2 mm, 3 mm, and 4 mm) were patterned into collagen to determine the smallest viable channel size. The mid-plane surfaces (Figure 3A) and cross-sections (Figure 3B) of the channels were imaged and analyzed, which showed that structurally intact channels in collagen could be made down to a 400 µm diameter with the current approach. To determine the effect of collagen density on the structural integrity of the patterned channel, we compared the midplane surface area of channels patterned into collagen using the 4 mm open mold at collagen densities of 2, 3, and 4 mg/mL (Figure 3C). The measurement at a collagen density of 0 mg/mL represents the midplane surface area of a pristine 3D-printed mold. There was no statistically significant difference for the tested collagen densities; however, the density of 4 mg/mL provided channels most similar to the theoretical mold value.

The midplane surface areas (Figure 3D) and cross-sectional areas (Figure 3E) of the patterned channels were quantified and compared to their respective 3D-printed molds. The midplane surface area and cross-sectional area for all channel sizes were not significantly different from their respective mold values (*p*-values > 0.05). These results indicate that this approach can effectively replicate intricate geometries in collagen down to a 400 µm diameter, exhibiting remarkable fidelity to the original 3D-printed mold designs across various collagen densities, thereby offering a robust platform for tissue modeling applications.

### 3.4. Formation and Characterization of Open Blood Vessel Mimics

To determine the suitability of the patterned open channels for generating blood vessel mimics, HUVECs were seeded into the 800 µm and 2 mm diameter channels. Briefly, 800 µm and 2 mm diameter semi-cylindrical molds were placed in 24-well plates, and 4 mg/mL collagen was dispensed around the mold and allowed to gel. The molds were removed, then seeded with HUVEC cells (Figure 4A). After five days from initial cell seeding, the blood vessel mimics were fluorescently stained and imaged under confocal microscopy. Both the 800 µm and 2 mm diameter blood vessel mimics exhibited physiological curvature with distinct vessel wall boundaries (cross-sectional views in Figure 4B,C). There was also uniform cell coverage throughout the total surface area of the open channel. Cell viability analysis of the 800 µm and 2 mm channels using AO/PI and fluorescent confocal imaging showed 89.3% and 90.4% viability, respectively (Figure 4D).

### 3.5. Effect of Mild Hypoxia on Open Blood Vessel Mimics

To demonstrate the suitability of the open blood vessel mimics for biological studies, they were used to simulate hypoxic vasculature, which is a condition that can manifest under disorders such as chronic obstructive pulmonary disease (COPD) or cystic fibrosis [29] and in certain environments (e.g., outer space [30,31]). Specifically, the 2 mm diameter open vessels were subjected to 16% $O_{2}$, representative of mild hypoxic conditions that would be experienced by astronauts, and a 20% $O_{2}$ (normoxia) control condition. Cultures in hypoxia had a significantly lower cell viability of 87.2% versus a viability of 95.5% in normoxia (**** *p*< 0.001, Figure 5A,B). The hypoxic culture also displayed lower CD31 surface area coverage of 24.9% compared to 32.3% in the normoxia culture (* *p* = 0.0110, Figure 5C,D).

## 4. Discussion

The most common methods of cell culture utilize 2D platforms, which often lack a physiologically relevant structure and microenvironmental cues. Open microfluidic models are emerging as a promising method to culture cells on platforms more representative of in vivo conditions without greatly increasing culture cost or fabrication time. We successfully created a simple and customizable method to create open microfluidic models with tissue-mimicking architecture in hydrogels. The 3D-printed molds created in this work can be printed on resin-based or extrusion-based 3D printers and have relatively low cost and time requirements compared to traditional fabrication methods. For example, soft lithography has previously been estimated to cost USD 55 per sample in materials, not including cleanroom or machine costs [32]. We estimate our material cost to approximately USD 1–USD 2.50 per sample depending on the type of 3D printing technology used to fabricate the molds.

The molds can be designed to any desired architecture using standard computer-aided design software, making them tailorable to many biological applications. In this study, we demonstrated hydrogel molding with star-shaped well, square well, round well, and open channel geometries. The approach is amenable to different types and densities of hydrogel, including agarose (1% (*w*/*v*) and 2.5% (*w*/*v*)), gelatin (5% (*w*/*v*) and 10% (*w*/*v*)), and collagen (2 mg/mL, 3 mg/mL and 4 mg/mL), which increases the number of applications this technique could be used for. Moreover, the dimensions of the molds can be adjusted to the required application. To demonstrate this capability, six different diameters for the open-channel molds were evaluated, ranging from 400 µm to 4 mm. Based on the measured midplane surface area and cross-sectional area of the open channels, the molds can create a pattern in collagen gel identical to the theoretical model values within statistical testing parameters. Open channels with diameters smaller than 400 µm were not obtainable using our current 3D printers. We were unable to create molds with reproducible curvature due to limitations with our printer’s resolution; however, smaller diameters may be achievable with next-generation models of the printers or using other additive manufacturing methods with higher resolution. In addition to channel diameter, we also evaluated the range of collagen densities that would be suitable for generating physiological architecture using the molds. All tested densities from 2 mg/mL to 4 mg/mL were suitable for channel formation, which demonstrates the potential for tailoring the mechanical properties of the collagen matrix, such as matrix stiffness.

As proof-of-demonstration of our approach for tissue modeling, blood vessel mimics were formed by seeding HUVECs into the 800 µm and 2 mm diameter open channels. Over the duration of the culture period, the blood vessel mimics exhibited growth and proliferation, forming a uniform endothelium on the patterned collagen. Fluorescent confocal images of the blood vessel mimics showed identifiable vessel walls and curvature, indicating closer recapitulation of curvature seen in vivo compared to traditional 2D culture methods [2]. Moreover, the blood vessel mimics had sufficient cell coverage and viability (90%) and expressed the common vascular differentiation marker CD31. These results are consistent with previous vessel-on-a-chip platforms [33,34] and show that the current approach can be used to produce engineered blood vessel mimics more representative than 2D cultures.

To further demonstrate the capability of our approach for real-world application, we used the open blood vessel mimics to study the effects of mild hypoxia on vasculature, a condition that often arises in common lung conditions and strenuous environments. Vessel mimics grown in the 2 mm diameter open channels were subjected to 16% $O_{2}$ (hypoxia) and 20% $O_{2}$ (normoxia). The cell viability of the blood vessel mimics under mild hypoxia was 8.3% lower than the viability measured in the normoxic culture. These findings are consistent with previous studies investigating the effects of hypoxia on HUVECs [35] or cells in engineered microvascular networks [36]. Furthermore, CD31 expression was 7.4% lower in the hypoxic culture, indicating that the blood vessel mimics may have visibly covered the curvature of the open channel, but had a percentage loss in their cell–cell junctions when compared to the normoxia condition. These results confirm findings from previous studies which also indicate lower CD31 expression in endothelial cells exposed to hypoxia [37,38,39]. Loss of CD31, which results in an increase in vascular permeability, can lead to potential health concerns, including inflammation, atherosclerosis, and drug resistance [40,41]. Collectively, the results of the current study showed that our approach can generate hydrogel models with tissue-mimicking architecture for in vitro studies in tailored microenvironments. These hydrogel models could be used for future clinical research investigating the effects of hypoxia in the cancer tumor microenvironment or the cellular response to inflammation or fibrosis. There are also potential drug delivery applications, including preclinical in vitro testing of hypoxia-targeted drug formulations [42].

Future work using these models could include performing additional studies to test the functionality of the vessel mimics, including cytokine secretion assays and testing barrier function. This study presents some limitations that should be considered. The proposed method can produce open channels with diameters of >400 µm, but this molding method does not currently allow for full lumens to be created. Although open channel-microfluidics are not as physiologically relevant for modeling microvessels as a closed lumen, they still present advantages of easier media collection and monitoring, as well as curvature that is more relevant than 2D cultures [22]. Further, scaling up microfluidic and 3D culture systems to an industrial scale for high-throughput screening or clinical applications presents significant challenges. Our system would need to address challenges associated with manufacturing and maintaining system functionality before being transitioned from the laboratory to a larger industrial scale.

## 5. Conclusions

In this paper, we developed a simple and customizable method for generating open microfluidic cell culture models using 3D-printed molds. We demonstrated the production of different model geometries and sizes using star-shaped well, square well, round well, and open-channel molds. We demonstrated that the molds can be used to generate architectures in collagen (2 mg/mL, 3 mg/mL and 4 mg/mL), agarose (1% (*w*/*v*) and 2.5% (*w*/*v*)), and gelatin (5% (*w*/*v*) and 10% (*w*/*v*)). All hydrogel architectures were structurally intact and suitable for cell culture applications. Open channels with diameters ranging from 400 µm to 4 mm were created with high fidelity compared to the 3D-printed mold designs. Blood vessel mimics were created by culturing HUVECs in open channels patterned in collagen, with characteristic vascular structure and cell junction expression, and sufficient viability (>89%). Moreover, the microenvironmental oxygen levels for the blood vessel mimics were adjusted to simulate mild hypoxia (16% $O_{2}$), showing decreased viability and CD31 expression. Overall, our approach enables the generation of open microfluidic tissue mimics that could have utility in basic biological studies and real-world applications.

## Figures and Tables

**Figure 1 bioengineering-12-00102-f001:**
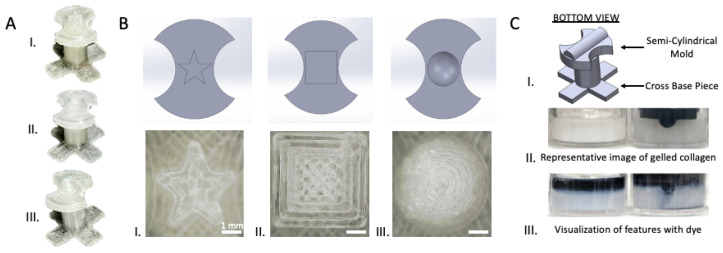
Three-dimensionally printed molds can be used to create various architectures in hydrogels. (**A**) Images of fully assembled printed molds in (**I**) star-shaped, (**II**) square and (**III**) round well shapes. (**B**) Top view of mold shapes to demonstrate ability to create different architectures in hydrogels. (**I**) Star-shaped CAD model and digital microscope image of star-shaped printed mold. (**II**) Square-well CAD model and digital microscope image of square-well printed mold. (**III**) Round-well CAD model and digital microscope image of round-well printed mold. (**C**) (**I**) SolidWorks model of fully assembled open-channel mold, showing bottom view of cross-base piece (**bottom**) and open channel (**top**). (**II**) Side view of gelled collagen without mold (**left** image) and with 3D-printed mold before mold removal (**right** image). (**III**) Side view of gelled collagen after mold has been removed, revealing open-channel architecture, versus gelled collagen with no molded shape. Dyed water was added for visualization purposes. All scale bars = 1 mm. Figure created with BioRender (https://www.biorender.com, accessed 11 December 2024).

**Figure 2 bioengineering-12-00102-f002:**
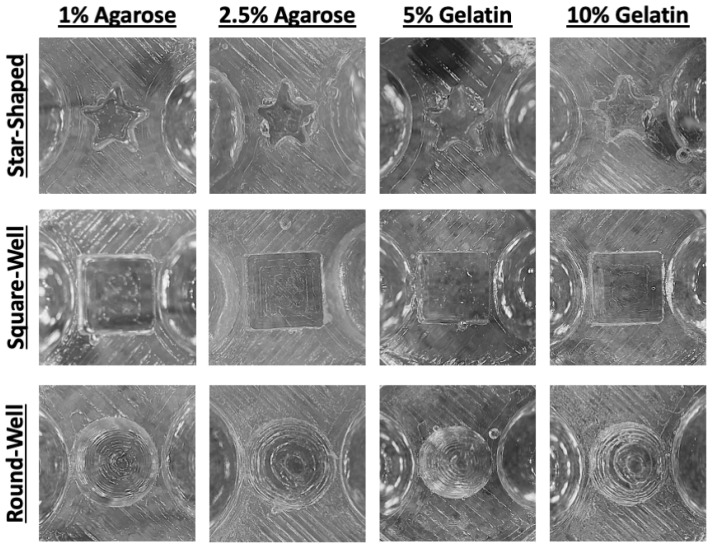
Various architectures, including star-shaped, square, and round well shapes, were molded into gelatin and agarose. Agarose [1% (*w*/*v*) and 2.5% (*w*/*v*)] and gelatin [5% (*w*/*v*) and 10% (*w*/*v*)] were successfully patterned with star-shaped, square, or round well shapes using the designed 3D-printed molds. The molded shapes were structurally intact at all of the tested hydrogel concentrations, demonstrating their potential utility for cell culture applications.

**Figure 3 bioengineering-12-00102-f003:**
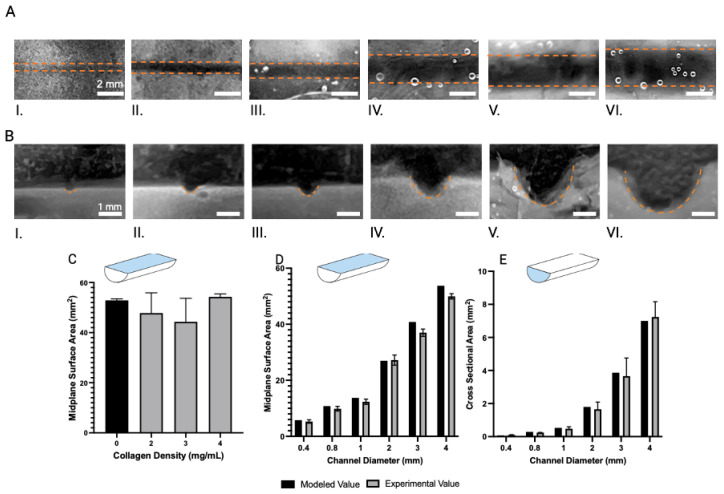
The experimental midplane and cross-sectional areas in collagen do not vary from the mold modeled value. Thus, this method is suitable for creating structurally intact open channels in collagen in a concentration ranging from 2 mg/mL to 4 mg/mL. (**A**) Representative images of the midplane surface area of various sizes of open channels (dashed line). (**I**) 400 µm, (**II**) 800 µm, (**III**) 1 mm, (**IV**) 2 mm, (**V**) 3 mm, and (**VI**) 4 mm channel diameters. (**B**) Representative images of the cross-sectional area of various sizes of open channels (orange dashed line). (**I**) 400 µm, (**II**) 800 µm, (**III**) 1 mm, (**IV**) 2 mm, (**V**) 3 mm, and (**VI**) 4 mm channel diameter. (**C**) Midplane surface area of 4 mm diameter channels at different collagen densities. A collagen density of zero represents the model value. ANOVA analysis indicated that there is no significant difference between the collagen concentration and surface area. (**D**) Comparing the midplane surface areas of the 3D-printed molds to the corresponding molded collagen at 4 mg/mL collagen density. (**E**) Comparing the cross-sectional area of 3D-printed molds to the corresponding molded collagen at 4 mg/mL collagen density. For (**C**–**E**), the error bars indicate the standard deviation, with n being at least three replicates. Dunnett’s test indicated no significance was found between the experimental values and the modeled control for any channel size or collagen density. The minimum sample size of *n* = 3 was used for each experiment.

**Figure 4 bioengineering-12-00102-f004:**
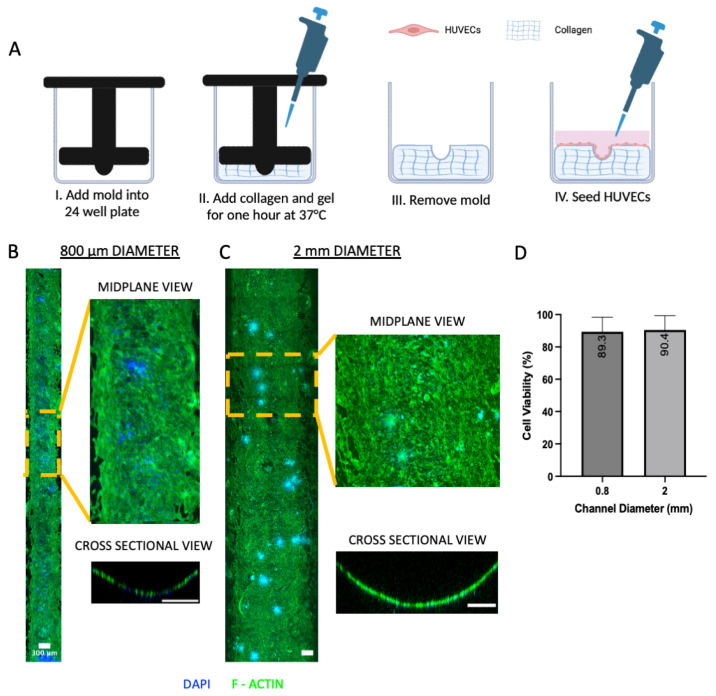
Blood vessel mimics were created by seeding human umbilical endothelial cells (HUVECs) in collagen open channels patterned with the 3D-printed semi-cylindrical molds. (**A**) Collagen molding scheme to create vessel mimics. (**I**) The assembled hydrogel mold was placed into a 24-well plate. (**II**) Collagen was dispersed through the crescent-shaped opening on the mold to fill the well plate area around the mold. (**III**) The mold was removed following gelation at room temperature for 15 min and incubation for one hour at 37 °C. (**IV**) HUVEC cells were seeded into the molded collagen architecture and allowed to culture at 37 °C. The entire midplane channel length, an exploded midplane view, and a cross-sectional view for the (**B**) 800 µm and (**C**) 2 mm channels are shown. HUVECs were fluorescently stained for F-Actin (AlexaFluor 488 Phallodin) and DAPI for visualization. (**D**) Cell viability was determined to be sufficient for each blood vessel mimic size using AO/PI and confocal microscopy, then counted in MATLAB. Cell viability for the 800 µm and 2 mm channels was 89.3% and 90.4%, respectively, where *n* = 13 for both groups.

**Figure 5 bioengineering-12-00102-f005:**
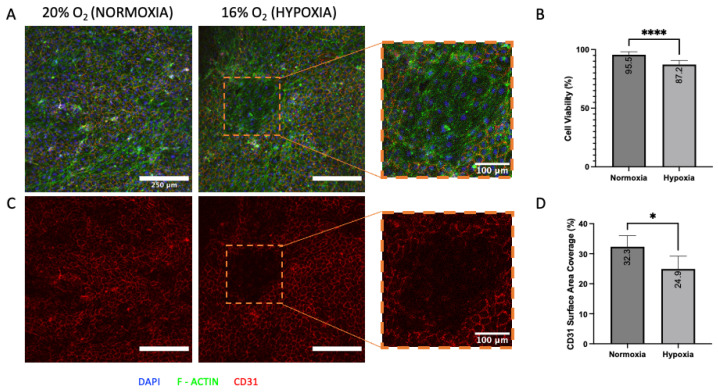
Open blood vessel mimics have lower cell viability and express less CD31 in mild hypoxic conditions. (**A**) Representative images of 20% oxygen (normoxia) versus 16% oxygen (hypoxia) culturing conditions with staining for CD31, F-Actin, and DAPI. (**B**) Cell viability of normoxic versus hypoxic conditions, where normoxia had a significantly higher viability compared to hypoxia (**** *p* < 0.001). (**C**) Representative images of CD31 only in both normoxia and hypoxia. (**D**) CD31 coverage in normoxia versus hypoxia revealed that significantly less CD31 was found in hypoxic conditions (* *p* = 0.0110). CD31 coverage was determined using threshold calculations in ImageJ Fiji 2. A sample size of *n* = 9 was used for both analyses.

## Data Availability

The original contributions presented in this study are included in the article. Further inquiries can be directed to the corresponding author.

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
