# Peer review of "Open Microfluidic Cell Culture in Hydrogels Enabled by 3D-Printed Molds"

_bioengineering, 2025, doi:10.3390/bioengineering12020102_

Round 1
Reviewer 1 Report
Comments and Suggestions for Authors
Review Report
This manuscript describes the preparation of 3D printed molds with different geometries, fabricated from agarose, gelatin, and collagen type I hydrogels. The various shapes include a star- shaped well, square-well, round-well, and open channel, which were able to demonstrate the versatility of this approach. To mimic blood vessels endothelial cells (HUVEC) were seeded in the open channels of the mold. Different channel diameters and collagen densities were tested. The newly formed blood vessels were exposed to hypoxia (mimicking pathological conditions) and the cell viability and CD31 surface expression were compared and analized in normoxia and hypoxia.
The basic idea of the paper is interesting, the methods are relatively easy and cheap to perform. However, before being published some corrections and modifications are needed.
When a material or an instrument is described you have to mention the manufacturer, the city and the country of fabrication or purchasing.
All abreviations used in the paper should be explained.
The description of the methodologies is not appropriate. You have to give all the details which is necessary to reproduce the process. How were the hydrogels dispensed in the wells? Pipetting? Was there any crosslinking technique used?
Line 127, how were the cell cultures fixed?
2.7 section, could you investigate the effect the different hydrogels on the cell viability? Were they biocompatible?
The various concentrations of collagen were studied. But I can not see any data about the optimization of HUVEC cell number for blood vessel creation. How could you selected the 600.000 cells/mL cell number? Can you provide any reference to this?
5-day cell maturation time was applied before microscopy and immunostainings. Was this period of time optimized previously?
Please show Fig4A more clearly by adding incubation times and temperature to the picture.
Please give reference to the sentence at line 281.
Please add this reference to the introduction:
Kocsis D, Kichou H, Döme K, Varga-Medveczky Z, Révész Z, Antal I, ErdÅ‘ F. Structural and Functional Analysis of Excised Skins and Human Reconstructed Epidermis with Confocal Raman Spectroscopy and in Microfluidic Diffusion Chambers. Pharmaceutics. 2022 Aug 13;14(8):1689. doi: 0.3390/pharmaceutics14081689.PMID: 36015315
Please give some outlook and explanation to the discussion and conclusion
what kind of functional studies could be performed for testing the functionality of the bioengineered new blood vessels?
Why have you prepared open channels? This is not physiologically relevant in case of microvessels. Did you have only technical issues behind this?
Are there any requirements concerning the physical properties of the hydrogels used for 3D printing? Please describe this.
Reviewer 2 Report
Comments and Suggestions for Authors
Review on Publication: Open Microfluidic Cell Culture in Hydrogels Enabled by 3D Printed Molds
The article presents an interesting topic and is prepared in a neat way, nevertheless it needs some corrections that the authors should take into account for the text to be published. My comments are as follows:
Please elaborate on potential clinical or industrial applications in the context of hypoxia research and drug delivery.
In the section on the manufacture of 3D printed moulds, the topic of the choice of PLA filament versus VeroBlue resin could be developed, taking into account the trade-offs between resolution, cost and material properties.
Explain in more detail why smaller channel diameters (<400 µm) could not be made with current technology and what future solutions could solve this problem.
Under the figures you should add short comments summarising the given result, please do not leave everything for discussion.
The conclusions are too short. They should be more elaborate and present the figures that the authors have reached in their research. In addition, please add a sentence about planned publications
Reviewer 3 Report
Comments and Suggestions for Authors
The article entitled “Open Microfluidic Cell Culture in Hydrogels Enabled by 3D Printed Molds” aims to develop a method to create open microfluidic cell cultures in vitro using 3D printed molds.
It is also important to ensure that the materials used are fully biocompatible and do not interfere with the cellular or molecular components of the tissue being modeled.
While the technology enables complex tissue structures, the actual utility of these complex models in standard research and drug testing has yet to be evaluated. A balance needs to be found between the complexity of the models and their practical applicability.
The additional complexity arising from the use of different hydrogel materials and structures should be justified by demonstrable benefits in terms of improved biological relevance or experimental results.
Scaling up microfluidic and 3D cell culture systems from the laboratory to industrial scale presents significant challenges, not only in manufacturing, but also in maintaining cell viability and system functionality over larger areas and volumes.
The economic and technical feasibility of scaling up these systems for high-throughput screening or clinical applications remains a question.
Although the systems aim to mimic in vivo conditions better than conventional 2D cultures, they cannot fully replicate the complex interactions and dynamics of living tissue, including immune responses, hormonal gradients and long-term aging effects.
The study focuses on specific cell types and conditions (such as hypoxia). The applicability of the results to other cells, tissues and diseases, especially those with multicellular interactions and chronic diseases, needs to be carefully assessed.
Round 2
Reviewer 3 Report
Comments and Suggestions for Authors
Accepted.